# Optimizing HCV Disease Prediction in Egypt: The hyOPTGB Framework

**DOI:** 10.3390/diagnostics13223439

**Published:** 2023-11-13

**Authors:** Ahmed M. Elshewey, Mahmoud Y. Shams, Sayed M. Tawfeek, Amal H. Alharbi, Abdelhameed Ibrahim, Abdelaziz A. Abdelhamid, Marwa M. Eid, Nima Khodadadi, Laith Abualigah, Doaa Sami Khafaga, Zahraa Tarek

**Affiliations:** 1Computer Science Department, Faculty of Computers and Information, Suez University, Suez 43533, Egypt; 2Faculty of Artificial Intelligence, Kafrelsheikh University, Kafrelsheikh 33516, Egypt; 3Department of Communications and Electronics, Delta Higher Institute of Engineering and Technology, Mansoura 35111, Egypt; 4Department of Computer Sciences, College of Computer and Information Sciences, Princess Nourah bint Abdulrahman University, P.O. Box 84428, Riyadh 11671, Saudi Arabia; 5Computer Engineering and Control Systems Department, Faculty of Engineering, Mansoura University, Mansoura 35516, Egypt; 6Department of Computer Science, Faculty of Computer and Information Sciences, Ain Shams University, Cairo 11566, Egypt; 7Department of Computer Science, College of Computing and Information Technology, Shaqra University, Shaqra 11961, Saudi Arabia; 8Faculty of Artificial Intelligence, Delta University for Science and Technology, Mansoura 35712, Egypt; 9Department of Civil and Architectural Engineering, University of Miami, Coral Gables, FL 33146, USA; nima.khodadadi@miami.edu; 10Computer Science Department, Prince Hussein Bin Abdullah Faculty for Information Technology, Al al-Bayt University, Mafraq 25113, Jordan; 11Department of Electrical and Computer Engineering, Lebanese American University, Byblos 13-5053, Lebanon; 12Hourani Center for Applied Scientific Research, Al-Ahliyya Amman University, Amman 19328, Jordan; 13MEU Research Unit, Middle East University, Amman 11831, Jordan; 14Applied Science Research Center, Applied Science Private University, Amman 11931, Jordan; 15School of Computer Sciences, Universiti Sains Malaysia, Gelugor 11800, Malaysia; 16School of Engineering and Technology, Sunway University Malaysia, Petaling Jaya 27500, Malaysia; 17Computer Science Department, Faculty of Computers and Information, Mansoura University, Mansoura 35561, Egypt

**Keywords:** hepatitis C virus (HCV), OPTUNA, hyperparameters, gradient boosting (GB), optimization

## Abstract

The paper focuses on the hepatitis C virus (HCV) infection in Egypt, which has one of the highest rates of HCV in the world. The high prevalence is linked to several factors, including the use of injection drugs, poor sterilization practices in medical facilities, and low public awareness. This paper introduces a hyOPTGB model, which employs an optimized gradient boosting (GB) classifier to predict HCV disease in Egypt. The model’s accuracy is enhanced by optimizing hyperparameters with the OPTUNA framework. Min-Max normalization is used as a preprocessing step for scaling the dataset values and using the forward selection (FS) wrapped method to identify essential features. The dataset used in the study contains 1385 instances and 29 features and is available at the UCI machine learning repository. The authors compare the performance of five machine learning models, including decision tree (DT), support vector machine (SVM), dummy classifier (DC), ridge classifier (RC), and bagging classifier (BC), with the hyOPTGB model. The system’s efficacy is assessed using various metrics, including accuracy, recall, precision, and F1-score. The hyOPTGB model outperformed the other machine learning models, achieving a 95.3% accuracy rate. The authors also compared the hyOPTGB model against other models proposed by authors who used the same dataset.

## 1. Introduction

Hepatitis C virus (HCV) is a blood-borne virus that primarily affects the liver, causing inflammation and potentially leading to cirrhosis, liver failure, and liver cancer [1,2]. HCV can be classified based on different factors, including the genotype, viral load, and disease stage. Genotype classification is based on the genetic makeup of the virus [3]. There are six major genotypes of HCV, each with several subtypes, and their distribution varies geographically. Genotype is essential in determining the course of treatment, as some genotypes are more responsive to certain medications than others. Viral load classification refers to the amount of virus in the blood [4]. High viral loads are associated with a greater risk of liver damage and may indicate a more aggressive disease course. Disease stage classification is based on the extent of liver damage caused by HCV. The disease can progress from a mild form of liver inflammation (chronic hepatitis) to more severe conditions of liver disease, such as cirrhosis and liver cancer. The stage of the disease is determined by evaluating liver function tests, imaging studies, and liver biopsy [5]. Overall, HCV disease classification plays a vital role in determining the appropriate course of treatment and monitoring disease progression. Early detection and treatment can help prevent or delay the development of severe liver complications associated with HCV infection [6]. In contrast to developed countries in Europe and North America, there is a higher incidence of hepatitis C virus (HCV) infection in the impoverished developing nations of Asia and Africa.

Moreover, countries like Pakistan, China, and Egypt have a larger population of individuals who suffer from chronic HCV infections [7,8]. The classification of HCV in Egypt is based on different factors, including genotype, viral load, and disease stage [7]. Genotype classification is based on the genetic makeup of the virus, and in Egypt, the most prevalent genotype is genotype 4. Viral load classification is based on the amount of virus present in the blood, and a high viral load is associated with a greater risk of liver damage. Disease stage classification is based on the extent of liver damage caused by HCV and can range from mild inflammation to severe liver diseases such as cirrhosis and liver cancer [9]. Healthcare workers in Egypt, who frequently interact with patients, are at an increased risk of HCV infection and other blood-borne diseases. Thus, it is crucial to understand the classification of HCV in Egypt to prevent and treat the disease effectively. Early detection and treatment of HCV can help prevent the development of severe liver complications associated with the disease [9].

There is a pressing need for accurate and dependable, non-invasive technology for the detection of HCV. Machine learning (ML) algorithms have effectively analyzed clinical data and identified intricate and nonlinear relationships in medical conditions. Classification techniques using ML algorithms can be employed to create models for diagnosing HCV by detecting infected individuals. However, unsuitable attributes in the attribute set can compromise the classifier’s accuracy [10,11]. Hyperparameter tuning is a crucial step in building a machine learning model. It involves selecting the optimal values for the hyperparameters that determine the algorithm’s behavior during training. Hyperparameters control various aspects of the model, such as the learning rate, regularization, number of layers, and number of nodes in each layer [12]. Choosing the right hyperparameters can significantly impact the model’s performance, accuracy, generalization, and convergence speed. An appropriate selection of hyperparameters can improve the model’s ability to generalize and make accurate predictions on unseen data. Conversely, inappropriate hyperparameters can lead to poor performance, overfitting, or underfitting. By optimizing the hyperparameters, we can improve the performance of the machine learning model, making it more accurate, robust, and efficient [13]. 

The term “optimization” refers to a vital instrument that is utilized in a variety of fields, one of which is the field of medicine, where it plays an important part. The goal of optimization is to arrive at the most advantageous results or choices attainable given a particular set of circumstances by making use of a variety of factors and predetermined standards. In the field of medicine, optimization is put to use in a wide variety of contexts, including the following: the prediction and classification of monkeypox disease [14,15,16], the feature selection and classification in diagnosed breast cancer [17], the classification of diabetes [18], and the classification of COVID-19 in chest X-ray pictures [19]. The application of optimization in the field of medicine helps to improve the overall outcomes for patients and makes the most efficient use of the available resources.

The dataset used in this study is available at the UCI machine learning repository [20]. The dataset includes 1385 instances and 29 features, where 28 are predictors, and one is the target feature. The multi-class target feature, representing the baseline histological staging, contains instances with different values. Specifically, the values are portal fibrosis without septa, few septa, many septa without cirrhosis, and cirrhosis—the number of instances associated with values of 336, 332, 355, and 362, respectively.

Feature selection is a critical step in building a machine learning model. It involves selecting the most relevant and informative features from the available dataset to train the model. The goal of feature selection is to reduce the dimensionality of the dataset, simplify the model, and improve its accuracy and generalization ability [21]. Feature selection is essential for several reasons. First, it can improve the accuracy and efficiency of the model. By removing irrelevant or redundant features, we can reduce the noise in the dataset, improve the model’s signal-to-noise ratio, and reduce overfitting. This, in turn, can improve the model’s performance on new, unseen data. Second, feature selection can simplify the model and make it more interpretable. By removing irrelevant or redundant features, we can reduce the complexity of the model and make it easier to understand and explain. This is particularly important in applications where model interpretability is critical, such as healthcare, finance, and legal domains. Third, feature selection can reduce the cost and time required to train the model. Reducing the number of features can decrease the computational resources needed for training, testing, and deploying the model [22,23]. Numerous studies have proposed various methods for predicting HCV disease, and we have developed our system, hyOPTGB, which utilizes the gradient boosting (GB) model and OPTUNA hyperparameter tuning to make these predictions [24,25]. The GB model has become increasingly popular for classification problems due to its strong performance [26,27].

Efficient hyperparameter tuning is essential to improve the performance of any system. A hyperparameter is a configuration setting that is not learned from the data but is set prior to the training of a machine learning model. These settings are essential because they control various aspects of the model’s training process and architecture, influencing its ability to learn and make predictions. Hyperparameters are distinct from model parameters, which are learned from the training data. Model parameters are the internal values that the machine learning algorithm adjusts during training to optimize the model’s performance for a specific task. A good set of hyperparameters can significantly impact the prediction system’s accuracy [28]. To achieve this, we chose to use OPTUNA for hyperparameter optimization, a widely used method that provides better optimization results. Each model has a unique set of hyperparameters and the goal is to find the best combination among them, a task known as hyperparameter optimization [29]. There are several ways to tackle this problem, including manual search, random search, grid search, and OPTUNA, which are considered the most effective. Manual tuning for a significant hyperparameter value is not recommended as it could be more efficient and effective [30]. Here are the key contributions of this paper:We presented a robust hyOPTGB system, which utilizes optimized GB and OPTUNA to predict HCV disease.The system’s performance was enhanced through data preprocessing techniques, feature selection, and hyperparameter tuning.GB is a distributed machine learning model based on trees renowned for its fast and efficient performance compared to other classification algorithms.OPTUNA is an automated technique used for hyperparameter tuning, which helped optimize our system’s performance.

The paper’s structure is organized as follows: Section 2 will provide an overview of the related work carried out by other researchers. Section 3 will elaborate on the materials and methods employed in this study. Section 4 will analyze the results obtained. Lastly, Section 5 and Section 6 will discuss and conclude the paper by summarizing the findings and proposing future research directions.

## 2. Related Work

Accurately determining the stage of liver fibrosis in patients with chronic hepatitis C (CHC) is critical for monitoring the disease, predicting treatment response, determining prognosis, and identifying the optimal treatment timing. However, a liver biopsy could be better due to its invasiveness, sampling errors, and high costs. To overcome these limitations, clinical information such as age, gender, body mass index (BMI), and non-invasive blood serum markers like alanine transaminase (ALT), aspartate transaminase (AST), glucose, hemoglobin (HGB), white blood cell (WBC) count, and red blood cell (RBC) count can be used to predict fibrosis stage. Various machine learning classification models have been applied in previous studies for HCV prediction, and this section discusses their results.

In a study by Tsvetkov et al. [24], a machine learning model was proposed to diagnose the stage of liver fibrosis in patients. The authors analyzed 1240 patient records with chronic viral hepatitis C and developed machine learning models using data from 689 patients classified by the stage of liver fibrosis. They identified critical predictors from nine widely used prognostic factors and obtained an accuracy of 80.56%. Akella et al. [25] conducted a study to create clinical risk models using machine learning algorithms to predict the extent of fibrosis in patients with chronic hepatitis C. They built nine ML algorithms using patient demographic information and standard serum laboratory values on an Egyptian cohort dataset. The extreme gradient boosting model achieved 81% precision in estimating fibrosis. The authors also found that most of their models performed better than existing diagnostic options in this patient group for assessing fibrosis.

The study conducted by Nandipati et al. [31] aimed to compare the performance of multiclass and binary class labels in predicting the hepatitis C virus using a dataset of Egyptian patients. They focused on identifying the essential features crucial in predicting the disease using different analytical tools. The results showed that a random forest using Python, and KNN using R had the highest precision rates of 54.56% and 51.06%, respectively, for both binary class and multiclass labels. Abd El-Salam et al. [32] used machine learning techniques to analyze a group of 4962 HCV patients in Egypt from 2006 to 2017. The study aimed to identify the presence or absence of esophageal varices in 2218 patients using 24 clinical laboratory variables. The researchers employed six popular classifiers: SVM, RF, C4.5, MLP, NB, and BN. The data were obtained from the Egyptian National Committee to Combat Viral Hepatitis, which manages the national treatment program for viral hepatitis patients in Egypt, overseen by the Ministry of Health. The study achieved accuracies ranging from 65.6% to 68.9% for the six classifiers.

Hashem et al. [33] utilized machine learning techniques to predict the development of hepatocellular carcinoma in patients with chronic liver disease due to HCV. They initially identified a group of input variables and then applied LR, DT, and CART algorithms to determine the optimal subset of variables. The findings revealed that the LR, DT, and CART algorithms achieved 96%, 99%, and 95.5% accuracy rates, respectively. Sartakhti et al. [34] proposed a new machine learning method that combines support vector machine (SVM) with simulated annealing (SA). They used a dataset from the UCI machine learning database and evaluated the classification accuracy using 10-fold cross-validation. The study found that the proposed approach achieved a high classification accuracy of 96.25%, surpassing other classification methods used for the same problem.

Vikas et al. [35] described a diagnostic system for identifying the presence of the hepatitis C virus (HCV). This system uses case-based reasoning (CBR) and correlation lift metrics to create a predictive model for detecting HCV. The proposed method can accurately predict whether a patient’s record shows HCV indications or not and can help reduce the risk factors associated with HCV in humans. Additionally, the system can differentiate between patient records of living and deceased individuals. Overall, this approach is highly effective in detecting HCV and can potentially decrease the virus’s prevalence in the population. Zaki et al. [36] discussed how a rough set of data were examined to identify attribute dependency and to create a reduced set of attributes. The primary objective of the dataset was to predict whether HCV was present or absent. The analysis revealed that the proposed methodology was exact and achieved high accuracy in predicting HCV’s presence or absence.

KayvanJoo et al. [37] conducted a study to predict the presence of HCV using machine learning techniques that analyze viral nucleotides. The research involved four approaches, DT, SVM, NB, and NN, to predict the response to interferon-alpha (IFN-alpha) and ribavirin (RBV) therapy based on processed features. The authors selected ten attribute weighting models from an initial dataset of 76 attributes, such as chi-square, Gini index, deviation, info-gain, info-gain ratio, SVM, PCA, uncertainty, relief, and rule. Finally, SVM, NB, NN, and DT were utilized for the classification process, with an average accuracy rate of 85%.

Until now, several methods have been employed to predict HCV disease prediction such as machine learning algorithms, feature selection methods, and deep learning methods. However, these methods have yet to be very effective in producing accurate predictions. To address this issue, this paper introduced a new model, hyOPTGB, which utilizes the GB model in combination with the trending hyperparameter optimization technique, OPTUNA. OPTUNA was chosen for its speed, efficiency, and proven success as one of the best available hyperparameter optimization techniques. Lai et al. [38] employed Optuna algorithm to optimize hyperparameters for forecasting models in this study. The findings of this investigation demonstrated that the utilization of Optuna in conjunction with five different tree-based machine learning models yielded highly satisfactory forecasting accuracy. Tonmoy et al. [39] introduced Optuna, an automated hyperparameter tuning algorithm, utilized to identify the best configurations for the dataset under investigation. The research introduces the architecture of the Optuna-Optimized GAN (OOG) method and showcases remarkable results, achieving accuracy, precision, recall, and F1 score.

Previous research studies failed to obtain good results due to their lack of hyperparameter optimization techniques. The hyOPTGB model aims to provide improved predictions for HCV disease prediction.

## 3. Materials and Methods

In this paper, we aim to improve the accuracy of predicting HCV disease among healthcare patients in Egypt by developing the hyOPTGB model. The system is designed to distinguish between HCV-infected patients and non-infected patients. Min-Max normalization is used to normalize the data; also, the forward selection (FS) wrapped method is used to select the most important features to increase the accuracy and efficiency of the classification process. To optimize the performance of the gradient boosting model, we fine-tuned its hyperparameters and then trained it using the resulting optimal parameters. We relied on the OPTUNA framework to carry out the hyperparameter tuning process. To evaluate the performance of our approach, we used a k-fold cross-validation (k = 15) method on the training data. Our optimization goal is to improve the model’s accuracy with each iteration. We have provided a visual representation of our proposed hyOPTGB model in Figure 1.

### 3.1. Dataset

The dataset used in this study is available at the UCI machine learning repository [20]. The dataset includes 1385 instances and 29 features, where 28 are predictors, and one is the target feature. The multi-class target feature, representing the baseline histological staging, contains instances with different values. Specifically, the values are portal fibrosis without septa, few septa, many septa without cirrhosis, and cirrhosis—the number of instances associated with values of 336, 332, 355, and 362, respectively. The description of the features is demonstrated in Table 1.

The heatmap analysis for the dataset features is given in Figure 2. The heatmap analysis is a valuable tool for visualizing the correlation matrix of a dataset, where each element in the matrix represents the correlation coefficient between two variables. In machine learning, a heatmap can be used to identify which features in a dataset are strongly correlated with the target variable and each other. Figure 3 displays the box plot for the dataset features. A box plot, also known as a box-and-whisker plot, is a graphical representation of the distribution of a dataset. It displays the median, quartiles, and outliers of the data. In machine learning, a box plot can be used to identify outliers, visualize the spread of the data, and compare the distributions of different features.

### 3.2. Min-Max Normalization

Min-Max normalization is a widely used data preprocessing method in machine learning that normalizes the values of a feature to a specific range (typically between 0 and 1) by subtracting the minimum value and dividing it by the range of the data [40]. This technique aims to ensure that all features are uniformly scaled and enhance the specific algorithm’s accuracy. The steps for Min-Max normalization are as follows:

For a given feature of x:Calculate the minimum value (min (x)) and maximum value (max (x)) of x across the dataset;Subtract the minimum value from each value of x (x − min (x));Divide the result by the range of the data (max (x) − min (x)).

### 3.3. Forward Selection

Forward selection is a feature selection technique in machine learning where features are added to the model iteratively based on their performance until a stopping criterion is met [41,42]. The basic algorithm for forward selection can be expressed in the following steps:Initialize the set of selected features S to be empty;For each feature X not in S, train a model with the features in S union X and compute the model’s performance;Select the feature X that yields the best performance on the model trained with S union X, and add it to the set S;Repeat steps 2 and 3 until a stopping criterion is met (e.g., a predetermined number of features have been selected, the performance improvement falls below a certain threshold, etc.).

Upon conducting forward selection on the dataset utilized in this study, a set of 13 features were chosen. These features include BMI, fever, diarrhea, epigastric pain, WBC, HGB, plat, ALT 1, ALT 4, ALT 24, base RNA, RNA 4, and RNA EF.

### 3.4. K-Fold Cross-Validation

K-fold cross-validation is a machine learning method used to assess a model’s performance with a small amount of data [43]. It works by splitting the data into k subsets of equal size or “folds”, using k − 1 of them to train the model and holding out the remaining fold for validation. This process is repeated k times, with each fold used once for validation and the remaining folds used for training the model [44]. 

The steps involved in k-fold cross-validation are as follows:Shuffle the dataset randomly;Split the dataset into k groups of equal size;For each group, select it as the validation set and use the remaining groups as the training set;Train the model using the training set and evaluate it using the validation set;Calculate the evaluation metric (e.g., accuracy, precision, recall, and F1-score.) for the model;Repeat steps 3–5 k times, using a different group as the validation set each time;Calculate the average evaluation metric over the k repetitions.

K-fold cross-validation helps to reduce the risk of overfitting the model to the training data by allowing the model to be tested on different subsets of the data. It also helps to reduce the variance of the evaluation metric since it uses multiple validation sets rather than a single one.

### 3.5. OPTUNA Optimization

To improve the performance of machine learning models, proper hyperparameter tuning is necessary. This step is crucial in creating effective models as it significantly impacts the model’s output. However, many individuals rely on trial and error, developing hyperparameters and testing them repeatedly over several hours or days. This approach could be more efficient and can be time-consuming. OPTUNA is a tool that automates the hyperparameter optimization process, eliminating the need for repetitive manual testing and saving valuable time and effort. OPTUNA is a tool that seeks to unify optimization paradigms by following a philosophy built upon four fundamental pillars.

#### 3.5.1. Design-by-Run API

The design-by-run API feature of OPTUNA enables the optimization of hyperparameters during the training process, eliminating the need for separate optimization runs and making the process more efficient. Design-by-run is a concept commonly used in deep learning to facilitate the dynamic programming of deep networks according to their intended function. However, in the context of optimization, this idea is utilized to create the search space dynamically. This means that design-by-run enables us to build the search space based on the optimization goals, allowing for a more flexible and customized approach to optimization [45]. OPTUNA defines hyperparameter optimization as optimizing an objective function by selecting an appropriate set of hyperparameters to achieve the best possible validation score [46]. To achieve this, OPTUNA employs a trailing object to construct the objective function and dynamically creates the search space as the objective function runs using the trail object’s methods. In other words, OPTUNA optimizes hyperparameters by iteratively refining the search-space based on the objective function’s output.

#### 3.5.2. Sampling

Sampling is a crucial part of the optimization process in OPTUNA, a hyperparameter optimization framework. OPTUNA supports both independent sampling and relational sampling [38]. In independent sampling, each hyperparameter is sampled independently of other hyperparameters. OPTUNA has several separate sampling algorithms, such as random search, grid search, and the Tree-structured Parzen Estimator (TPE). Relational sampling, on the other hand, considers the correlations between hyperparameters. OPTUNA supports relational sampling through the Covariance Matrix Adaptation Evolution Strategy (CMA-ES), a derivative-free optimization algorithm that can effectively explore high-dimensional and non-linear search spaces. The choice of sampling method in OPTUNA depends on the specific optimization task and the characteristics of the search space. Generally, independent sampling methods like TPE are more computationally efficient and work well for low-dimensional search spaces. In contrast, relational sampling methods like CMA-ES are better suited for high-dimensional search spaces with complex correlations between hyperparameters. OPTUNA also provides the flexibility to define custom sampling methods, allowing users to experiment with different sampling algorithms and tailor the optimization process to their needs.

#### 3.5.3. Pruning

Pruning is a technique used in hyperparameter optimization to reduce the computational resources needed for the optimization process [46]. OPTUNA provides several pruning algorithms to help users achieve faster and more efficient optimization. One of the most commonly used pruning algorithms in OPTUNA is successive halving, a type of early stopping technique [38]. In this method, the search space is split into multiple groups of hyperparameters, and each group is evaluated in parallel. The worst-performing half of the groups are eliminated, and the process is repeated until only one group of hyperparameters remains [39]. This reduces the number of trials needed to find the optimal hyperparameters. OPTUNA also supports asynchronous successive halving, which allows trials to run in parallel and dynamically allocates more resources to promising trials while cutting resources for less promising ones. Overall, pruning algorithms in OPTUNA help to reduce the computational cost of hyperparameter optimization while improving the efficiency and effectiveness of the search for optimal hyperparameters.

#### 3.5.4. Easy-to-Setup

By default, OPTUNA employs its memory data structure as a storage location; thus, making it effortless to use for simple purposes, which is an essential prerequisite for a contemporary hyperparameter optimization framework. OPTUNA offers numerous advanced capabilities, such as performing independent and relational sampling and providing various pruning algorithms, making it a superior framework for hyperparameter tuning. The architecture of OPTUNA is depicted in Figure 4. In OPTUNA, each worker in every sample executes the objective function (OF) once. The OF is executed through OPTUNA APIs, and when the moment API is called, it accesses the storage and retrieves any relevant data from previous samples stored in memory [39]. The workers operate independently, and they use the repository to track the results of the current study.

### 3.6. Gradient Boosting

Gradient boosting (GB) is a commonly used machine learning approach for classification and regression tasks. It is an ensemble learning method that combines multiple weak learners (basic models) into a stronger one [47,48]. The basic idea of gradient boosting is to iteratively add new vulnerable learners to the model, with each new learner trained to correct the mistakes made by the previous ones [49]. Gradient boosting typically uses decision trees to make predictions [50]. The loss-function quantifies the difference between predicted and actual output and is frequently utilized as the objective function for gradient boosting classifiers. The most commonly used loss functions for classification problems are cross-entropy and exponential loss. The cross-entropy loss is defined as given in Equation (1):(1)$$L\left(y,f\left(x\right)\right)=-\sum_{i=1}^{K}y_{i}\log\left(f_{i}\left(x\right)\right)$$
where $y_{i}\in \left[0,1\right]$ is the true label for class i, K is the number of classes, and $f_{i}\left(x\right)\in \left[0,1\right]$ is the predicted probability of class i. The exponential loss is defined as given in Equation (2):(2)L(y,f(x))=exp(−yf(x))
where y∈[−1,1] is the true label and f(x)∈ℝ is the predicted output of the model.

The objective function for gradient boosting is typically defined as the sum of the loss function over all the training examples, with the addition of a regularization term to prevent overfitting and given by Equation (3):(3)$$obj\left(\theta \right)=\sum_{i=1}^{n}L(y_{i},\sum_{j=1}^{m}f_{j}\left(x_{i},\theta \right))+\sum_{j=1}^{m}\Omega \left(f_{j}\right)$$
where n is the number of training examples, m is the number of trees in the ensemble, $f_{j}\left(x_{i},\theta \right)$ is the output of the j−th tree for the i−th training example, and $\Omega \left(f_{j}\right)$ is a regularization term that penalizes complex trees. The regularization term can take different forms, such as the L1 or L2 norm of the tree weights. The parameters θ of the objective function are the parameters of the trees in the ensemble.

The gradient boosting classifier aims to minimize the objective function with respect to the tree parameters θ using gradient descent. At each iteration, a new tree is added to the ensemble to reduce the residual error of the previous trees. The gradient of the objective function with respect to the model’s predicted output is used to train the new tree. The mathematical algorithm for gradient boosting can be described in Algorithm 1:
**Algorithm 1:** Gradient Boosting Classifier**Step 1:** Initialize the model by setting the initial predicted output of the model to be a constant value, such as the mean of the target variable.**Step 2:** For m=1 to M
(a)Compute the negative gradient of the loss function with respect to the predicted output of the model for each training example:
$$r_{im}=-{\frac{\partial L(y_{i},f\left(x_{i}\right))}{\partial f\left(x_{i}\right)}|}_{f\left(x\right)=f_{m-1}\left(x_{i}\right)}$$
where $L(y_{i},f\left(x_{i}\right)$ is the loss function, $y_{i}$ is the true label for the i−th training example, $f\left(x_{i}\right)$ is the predicted output of the model for the i−th training example, and $f_{m-1}\left(x_{i}\right)$ is the predicted output of the model up to the m−1−th iteration.(b)Fit a decision tree $h_{m}\left(x\right)$ to the negative gradient values $r_{im}$:
$$h_{m}\left(x\right)=argmin_{h}\overset{n}{\underset{i=1}{\sum }}{\left[r_{im}-h\left(x_{i}\right)\right]}^{2}$$
where h(x) is the output of the decision tree, and $h_{m}\left(x\right)$ is the output of the m−th decision tree.(c)Compute the step size $\gamma_{m}$ by minimizing the following objective function
$$\gamma_{m}=argmin_{\gamma }\overset{n}{\underset{i=1}{\sum }}L(y_{i},f_{m-1}\left(x_{i}\right)+\gamma h_{m}\left(x_{i}\right))$$
where $f_{m-1}\left(x_{i}\right)$ is the predicted output of the model up to the m−1−th iteration, and $f_{m}\left(x_{i}\right)=f_{m-1}\left(x_{i}\right)+\gamma_{m}h_{m}\left(x_{i}\right)$ is the predicted output of the model after the m−th iteration.(d)Update the predicted output of the model:
$$f_{m}\left(x\right)=f_{m-1}\left(x\right)+\gamma_{m}h_{m}\left(x\right)$$
where $f_{m-1}\left(x\right)$ is the predicted output of the model up to the m−1−th iteration, $h_{m}\left(x\right)$ is the output of the m−th decision tree, and $\gamma_{m}$ is the step size.
**Step 3:** Output the final predicted output of the model:
$$f_{M}\left(x\right)=\overset{M}{\underset{m=1}{\sum }}\gamma_{m}h_{m}\left(x\right)$$
where $f_{M}\left(x\right)$ is the final predicted output of the model, M is the number of iterations, $\gamma_{m}$ is the step size at the m−th iteration, and $h_{m}\left(x\right)$ is the output of the m−th decision tree.

### 3.7. Proposed hyOPTGB Model

After Min-Max normalization and feature selection, we utilized the OPTUNA optimization to optimize the hyperparameters of the gradient boosting model. We opted for eight specific hyperparameters: loss, learning_rate, n_estimators, subsample, criterion, min_samples_split, min_samples_leaf, and max_depth. Following the tuning process, we trained the gradient boosting model using the optimized hyperparameters. The hyperparameter called loss is the loss function that will be optimized, which is employed for binomial and multinomial deviance. This function is especially beneficial for classification tasks that require probabilistic outputs. Learning_rate reduces the impact of each tree during the training process. N_estimators determine the number of boosting rounds to execute in gradient boosting. Gradient boosting is typically resistant to overfitting, so increasing the number of rounds often leads to improved performance. Subsample refers to the proportion of the total training samples to be utilized for training each base learner. Criterion refers to the metric used for evaluating the effectiveness of a split. Min_samples_split is a hyperparameter that sets the minimum number of data points that must be present in a decision tree node to permit a split. Min_samples_leaf is a hyperparameter that defines the minimum number of instances that should be present in a leaf node of a tree. The Max_depth hyperparameter sets the maximum depth a tree can reach by limiting the number of nodes. Tuning this parameter is crucial for achieving optimal performance, and the ideal value depends on how the input features interact. If set to “none”, the tree will keep growing until all leaf nodes become pure or have fewer than the “min_samples_split” samples. Table 2 demonstrates the hyperparameters for the gradient boosting model using the default hyperparameters, and Table 3 illustrates the hyperparameters for the gradient boosting model using OPTUNA.

### 3.8. Machine Learning Classification Models

This section employs five different machine learning classification models, namely, decision tree (DT), support vector machine (SVM), dummy classifier (DC), ridge classifier (RC), and bagging classifier (BC), to evaluate and compare their performance with the proposed hyOPTGB model. Min-Max normalization and 15-fold cross-validation are performed for the five machine learning models using their default hyperparameters. The architecture for the five machine learning models is in Figure 5.

#### 3.8.1. Decision Tree

A decision tree (DT) is a machine learning algorithm that builds a tree-like model of decisions and their possible consequences [51]. It is a popular approach for solving classification problems in which the input data is classified into multiple classes based on a set of rules derived from the training data. The decision tree algorithm recursively splits the data into subsets based on the features that best discriminate between the classes until a stopping criterion is met [52]. The resulting tree can classify new data by following the path from the root to a leaf node corresponding to the predicted class.

#### 3.8.2. Support Vector Machine

Support vector machine (SVM) is a classification algorithm used in machine learning. It finds the optimal hyperplane that separates different classes in the input data [53]. In a two-class classification problem, SVM chooses the hyperplane that maximizes the margin between the two closest data points from different classes. The hyperplane is defined by the support vectors, which are the data points lying on the margin. SVMs can also handle non-linearly separable data by mapping it to a higher dimensional space using a kernel function [54]. This allows SVM to find the optimal hyperplane in the transformed space. SVMs are suitable for high-dimensional datasets and have widespread applications.

#### 3.8.3. Dummy Classifier

A dummy classifier (DC) is a classification algorithm used as a baseline model to compare the performance of other more advanced classifiers [55]. It is a simple algorithm that makes predictions by following a pre-defined rule, such as always predicting the most frequent class in the training data or randomly selecting a class based on the class distribution. A dummy classifier is used to evaluate whether a more complex classifier can outperform this basic algorithm [56]. Dummy classifiers are helpful in situations where the class distribution is imbalanced or when the performance of a classifier needs to be compared against a simple baseline.

#### 3.8.4. Ridge Classifier

A Ridge classifier (RC) is a linear classification algorithm that uses ridge regression to classify input data into multiple classes [57]. It works by finding the linear decision boundary that separates the different classes while minimizing the sum of the squared weights of the features. The regularization parameter, the ridge penalty, is added to the objective function to prevent overfitting of the model on the training data [58]. The ridge classifier is a variant of logistic regression that is particularly useful when dealing with high-dimensional datasets.

#### 3.8.5. Bagging Classifier

A bagging classifier (BC) is an ensemble learning algorithm that combines multiple base classifiers to improve the overall classification performance [59]. It works by generating multiple bootstrap samples of the training data, and each sample is used to train a different base classifier. The base classifiers can be any type of classifier, such as decision trees or SVMs [60]. During the prediction phase, the bagging classifier aggregates the predictions of all base classifiers using a majority voting scheme to make the final prediction. Bagging reduces the model’s variance and improves its generalization performance, especially when dealing with unstable base classifiers.

## 4. Results and Discussion

The Jupyter Notebook version 6.4.6 was utilized to carry out the experiments. It is a tool commonly employed for data analysis and visualization using Python. Jupyter Notebook offers a comprehensive environment to write, run, and document code and results, in addition to generating visual representations of data. This tool supports several programming languages, Python 3.8 included, and runs on a web browser. The experiments were executed on a computer that operates on Microsoft Windows 10, with an Intel Core i5 CPU and 16 GB of RAM. To assess the effectiveness of our models, we employed several metrics, including accuracy, recall, precision, and F1-score [61,62]. These metrics were used to evaluate and measure the model’s performance in different aspects, such as correctly identifying true positives (TP), true negatives (TN), false positives (FP), and false negatives (FN).
(4)$$\mathrm{Accuracy}=\frac{\mathrm{TP}+\mathrm{TN}}{\mathrm{TP}+\mathrm{FP}+\mathrm{FN}+\mathrm{TN}}$$
(5)$$\mathrm{Recall}=\frac{\mathrm{TP}}{\mathrm{TP}+\mathrm{FN}}$$
(6)$$\mathrm{Precision}=\frac{\mathrm{TP}}{\mathrm{TP}+\mathrm{FP}}$$
(7)$$F1\text{-}\mathrm{score}=\frac{2*\mathrm{Recall}*\mathrm{Precision}}{\mathrm{Recall}+\mathrm{Precision}}$$

Our machine learning classifier for the model was the gradient boosting model. Initially, we optimized the hyperparameters of the gradient boosting model using the OPTUNA framework. Once the hyperparameters were tuned, we used them to analyze the model’s performance. OPTUNA used a distinct set of hyperparameters for the gradient boosting model during the tuning phase, which helped enhance performance on 15-fold cross-validation. Also, Min-Max normalization and forward selection were applied to the dataset. These features selected by forward selection include BMI, fever, diarrhea, epigastric pain, WBC, HGB, plat, ALT 1, ALT 4, ALT 24, base RNA, RNA 4, and RNA EF. We assessed and contrasted the performance of five machine learning classification models, specifically the decision tree (DT), support vector machine (SVM), dummy classifier (DC), ridge classifier (RC), and bagging classifier (BC), against the proposed hyOPTGB model. We used Min-Max normalization and 15-fold cross-validation for the five machine learning models to conduct this evaluation using their default hyperparameters. Table 4 displays the accuracy, F1-score, recall, and precision results of the five machine learning models and the proposed hyOPTGB model for performance evaluation.

As seen in Table 4, the hyOPTGB model outperformed all other models, achieving an accuracy of 95.3%, F1-score of 94.8%, recall of 94.5%, and precision of 94.1%. The RC model comes in second place, with high metrics scores. Its accuracy, F1-score, recall, and precision are 84.6%, 84.4%, 83.7%, and 83.3%, respectively. The SVM model comes in third place; its accuracy, F1-score, recall, and precision are 84.4%, 84.1%, 83.5%, and 83.2%, respectively. The SVM model comes in fourth place; its accuracy, F1-score, recall, and precision are 83.9%, 83.5%, 83.2%, and 82.9%, respectively. The accuracy, F1-score, recall, and precision are 83.6%, 83.2%, 82.8%, and 82.5%, respectively, for the BC model, which comes in fifth place. The DC model obtains the worst results; its accuracy, F1-score, recall, and precision are 81.3%, 80.7%, 80.4%, and 80.2%, respectively. Figure 6 displays the accuracy of the five classification models and the proposed hyOPTGB model.

To ensure the strength and validity of the proposed model, we applied 10-fold cross-validation and 20-fold cross-validation on the hyOPTGB model and compared the results when using 15-fold cross-validation. Table 5 demonstrates the performance of the proposed hyOPTGB model using 10-fold cross-validation, 15-fold cross-validation, and 20-fold cross-validation, respectively.

As seen in Table 5, the best results are obtained when (k = 15); its accuracy, F1-score, recall, and precision are 95.3%, 94.8%, 94.5%, and 94.1%, respectively. The worst results are obtained when (k = 20); its accuracy, F1-score, recall, and precision are 91.2%, 90.9%, 90.6%, and 90.2%, respectively. The accuracy, F1-score, recall, and precision are 93.5%, 93.1%, 92.8%, and 92.7%, respectively, when (k = 10). Figure 7 demonstrates the accuracy percentage for the proposed hyOPTGB model using 10-fold cross-validation, 15-fold cross-validation, and 20-fold cross-validation, respectively. Table 6 displays a comparative study that used the same dataset in this paper.

As shown in Table 6, the proposed hyOPTGB model achieved better performance in terms of accuracy than the previous study. Figure 8 displays the accuracy percentage for the proposed hyOPTGB model, and previous studies used the same dataset.

When evaluating the results of the hyOPTGB model, accuracy is the criterion that is applied. Table 7 presents the results of twenty separate iterations of the hyOPTGB model. There is information supplied regarding these runs’ minimal, median, maximum, and mean accuracy. These data, taken from a large number of different model runs, enable an evaluation of the consistency and efficiency of the hyOPTGB model. Because of this explanation’s level of depth, you are able to evaluate the performance and dependability of the model.

The results of the hyOPTGB and comparison model ANOVA are presented in Table 8. The purpose of this statistical study is to investigate and explain model differences. The results of an ANOVA can demonstrate whether or not there is statistical variation in model performance. The Wilcoxon signed-rank test is used to make a comparison between the hyOPTGB model and the models that are being tested in Table 9. The results of this non-parametric test are compared based on matching data, such as how well a model performed on the same dataset. The Wilcoxon signed-rank test as well as ten distinct iterations of each model make it possible to make accurate comparisons, which in turn increases the reliability of the study. The hyOPTGB model is evaluated in an objective manner using these statistical tests, in comparison to the other models. The analysis of variance (ANOVA) and the Wilcoxon signed-rank test (Wilcoxon *p*-values) are both useful tools for determining whether or not differences in model performance are statistically significant. These findings help explain why the hyOPTGB model is more effective than other models and why it can be applied to this particular job or dataset.

The comparison of the hyOPTGB model with other accuracy-based models in Figure 9 provides a valuable visual representation of each model’s performance in terms of accuracy. This comparison can offer several insights and benefits:Model Performance Assessment: The plot allows readers to quickly assess the performance of various models in a single view. This is particularly important when evaluating machine learning models, as accuracy is a common metric for measuring predictive performance.Model Ranking: By plotting the accuracy achieved by each model, it becomes apparent which model outperforms the others and to what extent. This ranking can help identify the most effective model for the specific task or dataset under consideration.Identification of the Best Model: It aids in the identification of the best-performing model, which can be crucial for decision-making in practical applications.Comparative Analysis: The plot enables a side-by-side comparison of the hyOPTGB model with other models, hyOPTGB model excels in terms of accuracy.Validation of Results: Plots of model accuracy provide visual evidence of the research findings, making it easier for readers and reviewers to validate the reported results and conclusions.

The histograms of accuracy values displayed in Figure 10 provide a valuable visual representation of the dispersion and concentration of the model performance results, both for the hyOPTGB model and the comparative models. Histograms are effective tools for understanding the distribution of accuracy values. They reveal how the performance of each model is dispersed across different accuracy levels. This can help identify the range and variability of the results.

The inclusion of residual plots, homoscedasticity plots, QQ (Quantile–Quantile) plots, and heat maps for the hyOPTGB model and comparative models in Figure 11 offers a comprehensive assessment of the model performance and various aspects of the models. Here’s why these visualizations are significant:Residual Analysis: Residual plots allow for the examination of the differences between observed values and model predictions. They help identify patterns or deviations in the model’s errors, such as systematic biases or outliers.
Figure 11The residual plots, homoscedasticity plots, QQ plots and the heat maps used to compare and display the presented hyOPTGB model to other models.
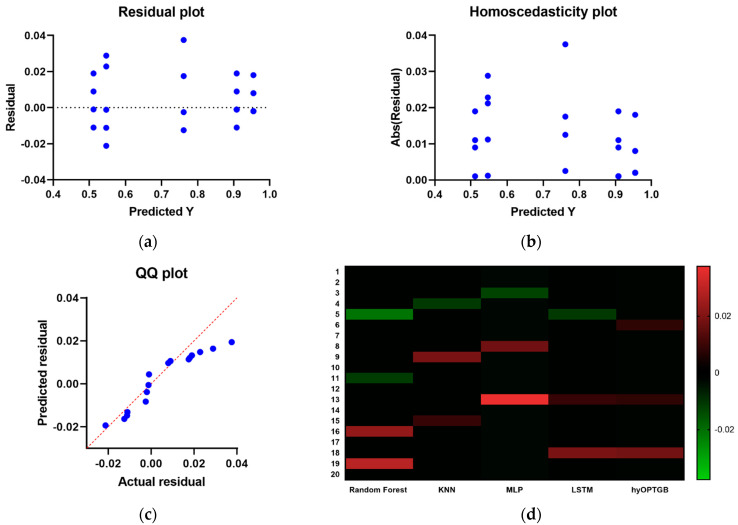

Homoscedasticity Assessment: Homoscedasticity plots are used to check the constant variance of model residuals across different levels of the predictor variable. They help ensure that the model’s errors have consistent variance and do not exhibit a trend.QQ Plots for Normality: QQ plots are particularly useful for assessing whether the residuals follow a normal distribution. Deviations from the expected straight line in a QQ plot can indicate departures from normality in the residuals.Heat Maps for Correlation: Heat maps provide a visual representation of the correlation between variables. In the context of machine learning, they can illustrate the correlation between features, residuals, or model predictions, helping to identify multicollinearity or patterns in the data.

The comparison of ROC curves presented in Figure 12 is a crucial element in evaluating and understanding the performance of binary classification models, specifically the hy-OPTGB model and the LSTM model. Here’s why these ROC curves are significant:Assessment of Binary Classification: ROC curves are a standard tool for assessing the performance of binary classification models. They help to understand how well a model can discriminate between the positive class and the negative class by varying the classification threshold.Sensitivity and Specificity Trade-off: The ROC curve graphically depicts the trade-off between sensitivity (the true positive rate) and specificity (the true negative rate) as the threshold for classification changes. It shows how different decision criteria can affect the balance between correctly identifying positive cases (sensitivity) and correctly identifying negative cases (specificity).Model Comparison: By comparing the ROC curves of two different models, such as the hyOPTGB model and the LSTM model, one can assess and visually contrast their classification performance. This comparison can provide insights into which model performs better at different operating points and threshold settings.Area Under the Curve (AUC): The area under the ROC curve (AUC) is often used as a single summary measure of a model’s overall performance. A higher AUC indicates a better model, as it represents a larger area under the ROC curve and suggests a better balance between sensitivity and specificity.

## 5. Discussion

The novelty of the study can be summarized in several key aspects:Targeted Research Area: The paper focuses on hepatitis C virus (HCV) infection in Egypt, a region with one of the highest HCV prevalence rates globally. This specific geographic context is significant, as it addresses a pressing health issue in a region with distinct risk factors and challenges.Unique Dataset: The study employs a dataset containing 1385 instances and 29 features related to HCV in Egypt, sourced from the UCI machine learning repository. This dataset is a valuable resource and forms the basis of the research, contributing to the understanding of HCV in this context.Development of hyOPTGB Model: A novel predictive model called hyOPTGB, which leverages an optimized gradient boosting (GB) classifier. The novelty here lies in the development of this specific model tailored to predict HCV disease in Egypt. The optimization of hyperparameters with the OPTUNA framework further enhances the model’s performance.Feature Selection and Preprocessing: The paper applies Min-Max normalization as a preprocessing step for scaling the dataset values and utilizes the forward selection (FS) wrapped method for identifying essential features. These techniques are integral to the model’s performance and contribute to the overall methodology’s uniqueness.Comparative Analysis: This paper conducts a comprehensive comparison of the hyOPTGB model against five other machine learning models, including decision tree (DT), support vector machine (SVM), dummy classifier (DC), ridge classifier (RC), and bagging classifier (BC). This comparative analysis helps demonstrate the superiority of their proposed model.High Accuracy Rate: One of the key findings is that the hyOPTGB model outperformed the other machine learning models with a remarkable 95.3% accuracy rate. Such a high accuracy rate in the prediction of HCV in Egypt is a notable contribution, indicating the efficacy of their model.Comparison with Other Studies: The paper also compares the hyOPTGB model with models proposed by different authors who used the same dataset. This comparative analysis helps establish the superiority of their model within the context of existing research.

## 6. Conclusions and Future Work

This research paper presents hyOPTGB, a model that employs an optimized gradient boosting (GB) classifier to predict HCV disease in Egypt. The model’s accuracy is enhanced using hyperparameter tuning with the OPTUNA framework, while the essential features in the dataset are identified using the forward selection (FS) wrapped method. Also, the Min-Max normalization preprocessing technique is used to scale the values of a dataset to a fixed range. The UCI machine learning repository provided the dataset containing 1385 instances and 29 features. The study compares hyOPTGB with five other machine learning models, namely, decision tree (DT), support vector machine (SVM), dummy classifier (DC), ridge classifier (RC), and bagging classifier (BC). It evaluates their efficiency using accuracy, recall, precision, and F1-score. The five machine learning models used their default hyperparameters. The hyOPTGB model outperforms the other machine learning models, achieving a 95.3% accuracy. The paper also conducts a comparative study of the proposed hyOPTGB model against those used by other researchers who employed the same dataset, and the results depicted that the proposed model achieved the best results. Some potential future directions for HCV disease prediction can be conducted in the future as (1) using genetic information; this can inform the development of models that incorporate genetic information to predict HCV infection risk. This approach can help identify genetic markers associated with HCV susceptibility, which can inform targeted prevention and intervention efforts. (2) Integrating multiple data sources by combining data from multiple sources, such as electronic health records, public health surveillance data, and social media, can improve the accuracy of predictive models and identify novel risk factors for HCV infection.

## Figures and Tables

**Figure 1 diagnostics-13-03439-f001:**
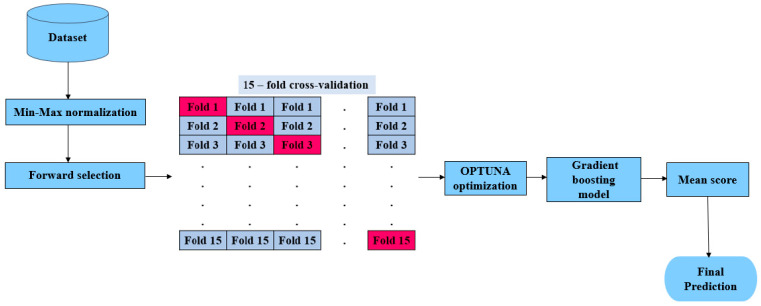
Representation of hyOPTGB model.

**Figure 2 diagnostics-13-03439-f002:**
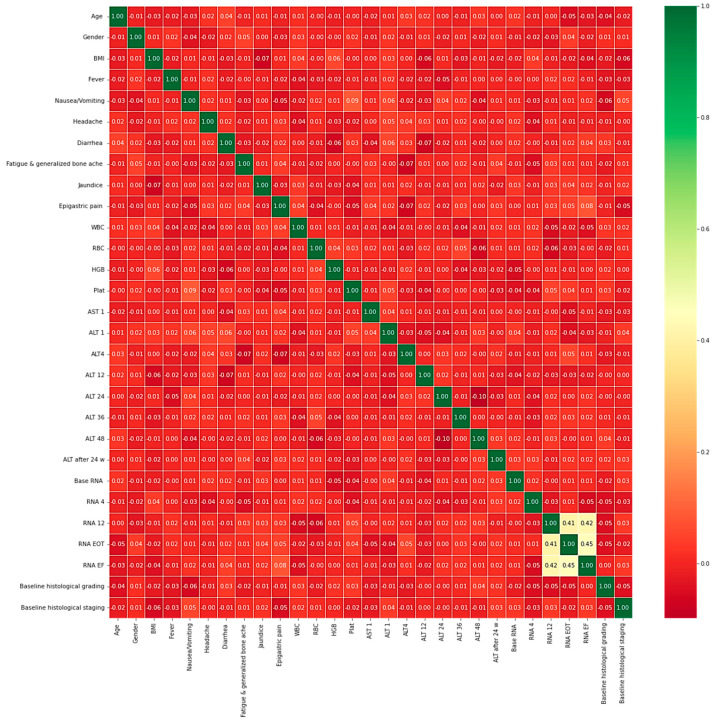
Heatmap analysis for the dataset features.

**Figure 3 diagnostics-13-03439-f003:**
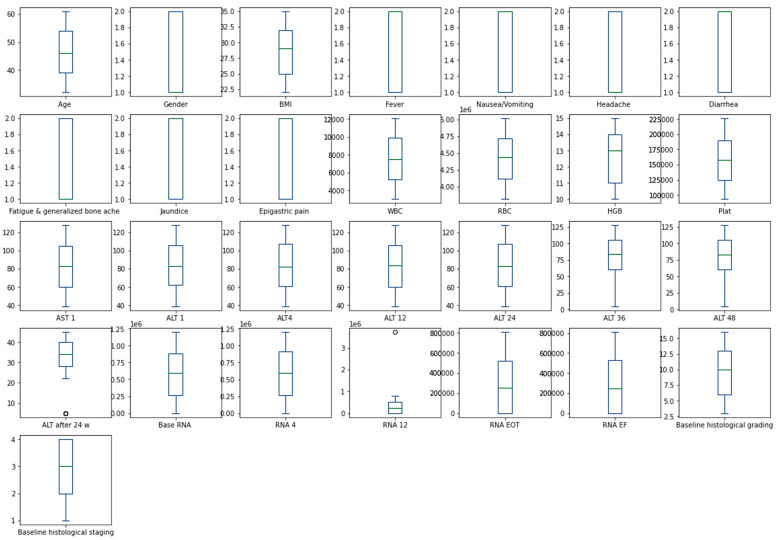
Box plot for dataset features.

**Figure 4 diagnostics-13-03439-f004:**
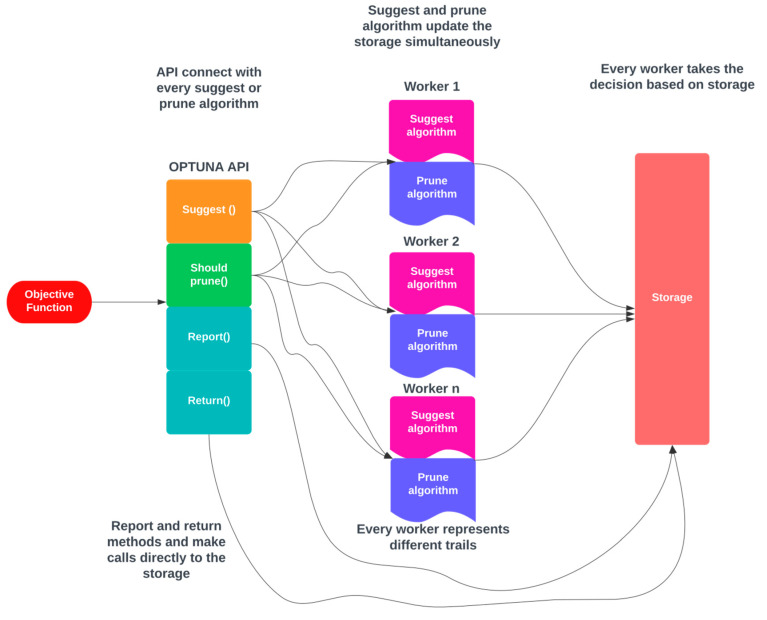
Architecture for OPTUNA.

**Figure 5 diagnostics-13-03439-f005:**
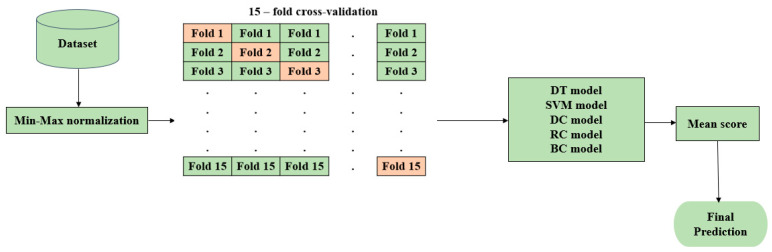
Architecture for the five machine learning classification models.

**Figure 6 diagnostics-13-03439-f006:**
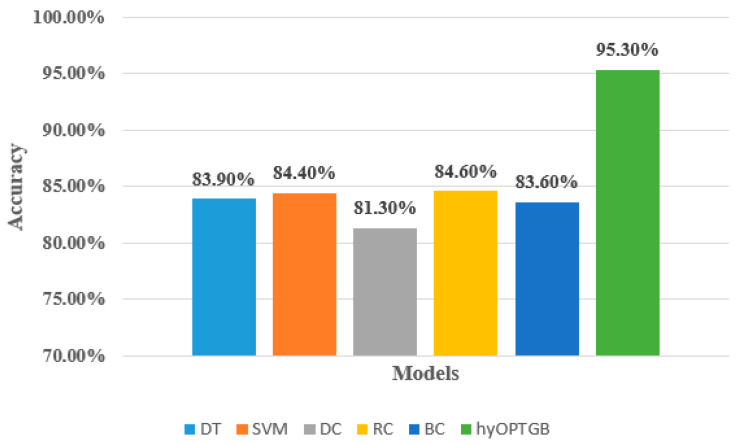
Comparison between the proposed hyOPTGB model and different machine learning models in terms of accuracy.

**Figure 7 diagnostics-13-03439-f007:**
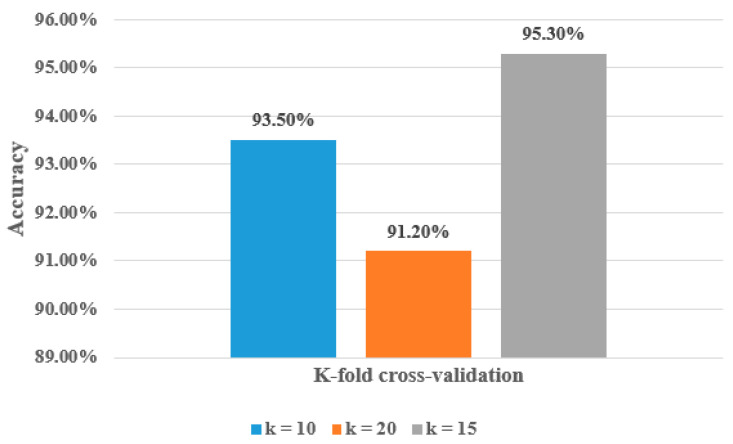
Accuracy percentage for the proposed hyOPTGB model using 10-fold cross-validation, 15-fold cross-validation, and 20-fold cross-validation in terms of accuracy.

**Figure 8 diagnostics-13-03439-f008:**
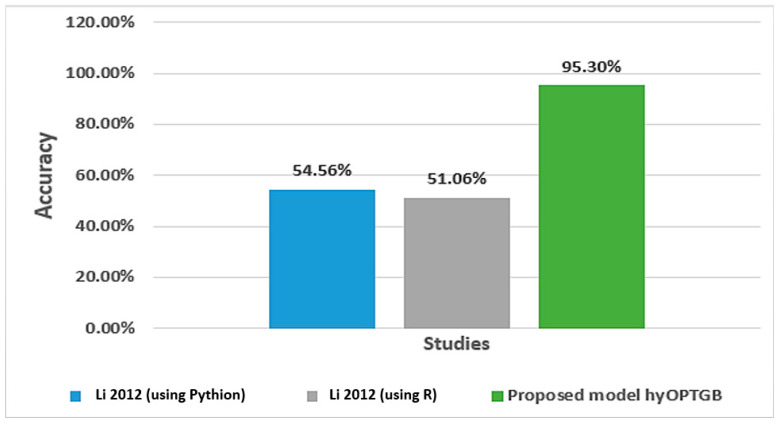
Accuracy percentage for the proposed hyOPTGB model, and previous studies used the same dataset [27].

**Figure 9 diagnostics-13-03439-f009:**
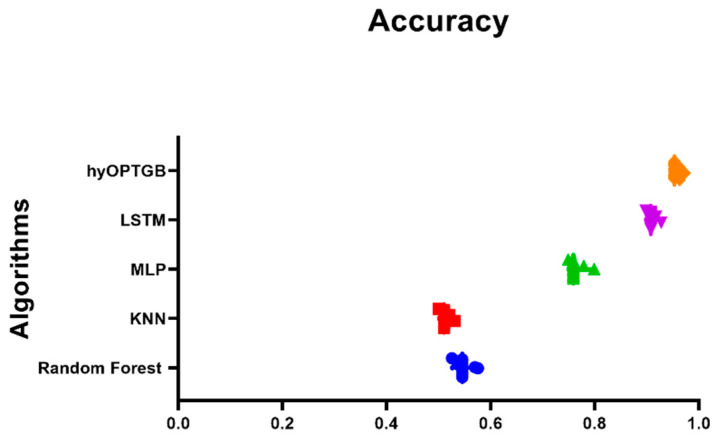
Plot of the presented hyOPTGB model and other models based on the accuracy.

**Figure 10 diagnostics-13-03439-f010:**
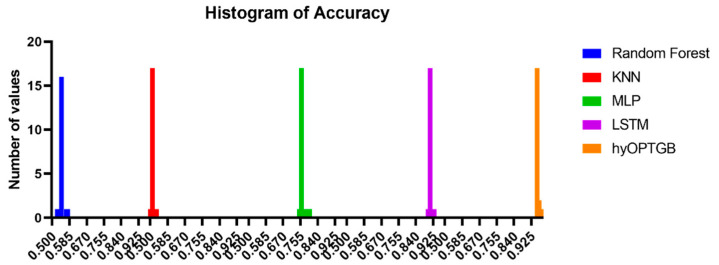
Histogram of accuracy for the presented hyOPTGB model and other models.

**Figure 12 diagnostics-13-03439-f012:**
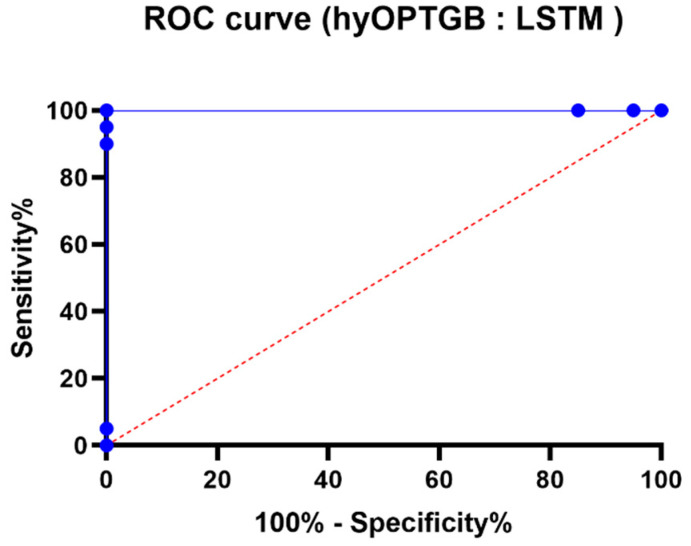
The ROC curve of the presented hyOPTGB model and LSTM model.

**Table 1 diagnostics-13-03439-t001:** Features description.

No.	Description	No.	Description
1	Age	16	ALT 1 (Alanine transaminase in 1 week)
2	Gender	17	ALT 4 (Alanine transaminase in 4 weeks)
3	BMI (Body mass index)	18	ALT 12 (Alanine transaminase in 12 weeks)
4	Fever	19	ALT 24 (Alanine transaminase in 24 weeks)
5	Nausea/Vomiting	20	ALT 36 (Alanine transaminase in 36 weeks)
6	Headache	21	ALT 48 (Alanine transaminase in 48 weeks)
7	Diarrhea	22	ALT after 24 w (Alanine transaminase after 24 weeks)
8	Fatigue & generalized bone ache	23	Base RNA (Ribonucleic acid)
9	Jaundice	24	RNA 4 (Ribonucleic acid in 4 weeks)
10	Epigastric pain	25	RNA 12 (Ribonucleic acid in 12 weeks)
11	WBC (White blood cell)	26	RNA EOT (Ribonucleic acid end of treatment)
12	RBC (Red blood cell)	27	RNA EF (Ribonucleic acid elongation factor)
13	HGB (Hemoglobin)	28	Baseline histological grading
14	Plat (Platelets)	29	Baseline histological staging (target feature)
15	AST 1 (Aspartate transaminase)		

**Table 2 diagnostics-13-03439-t002:** Default hyperparameters for the gradient boosting model.

No.	Parameters	Default
1	Loss	Log_loss
2	Learning_rate	0.1
3	N_estimators	100
4	Subsample	1
5	Criterion	Friedman_mse
6	Min_samples_split	2
7	Min_samples_leaf	1
8	Max_depth	3

**Table 3 diagnostics-13-03439-t003:** Optimized hyperparameters for the gradient boosting model using OPTUNA.

No.	Parameters	Default
1	Loss	Deviance
2	Learning_rate	0.7
3	N_estimators	582
4	Subsample	0.4
5	Criterion	Squared_error
6	Min_samples_split	9
7	Min_samples_leaf	7
8	Max_depth	12

**Table 4 diagnostics-13-03439-t004:** Performance of the five classification models and the proposed hyOPTGB model.

Models	Accuracy	F1-Score	Recall	Precision
**DT**	83.9%	83.5%	83.2%	82.9%
**SVM**	84.4%	84.1%	83.5%	83.2%
**DC**	81.3%	80.7%	80.4%	80.2%
**RC**	84.6%	84.4%	83.7%	83.3%
**BC**	83.6%	83.2%	82.8%	82.5%
**hyOPTGB**	**95.3%**	**94.8%**	**94.5%**	**94.1%**

**Table 5 diagnostics-13-03439-t005:** Performance of the proposed hyOPTGB model using different k-fold cross-validation.

K-Fold Cross-Validation	Accuracy	F1-Score	Recall	Precision
**k = 10**	93.5%	93.1%	92.8%	92.7%
**k = 15**	**95.3%**	**94.8%**	**94.5%**	**94.1%**
**k = 20**	91.2%	90.9%	90.6%	90.2%

**Table 6 diagnostics-13-03439-t006:** Comparative study of this work with another study used the same dataset.

Studies	Model	Accuracy
Ref. [27]	Random Forest Using Python	54.56%
Ref. [27]	KNN Using R	51.06%
Proposed model hyOPTGB	Gradient boosting using OPTUNA	**95.3%**

**Table 7 diagnostics-13-03439-t007:** A description of the hyOPTGB model that was proposed as well as the results of other models based on the accuracy factor.

	Random Forest	KNN	MLP	LSTM	hyOPTGB
Number of values	20	20	20	20	20
Minimum	0.5256	0.5006	0.7489	0.8976	0.953
25% Percentile	0.5456	0.5106	0.7589	0.9076	0.953
Median	0.5456	0.5106	0.7589	0.9076	0.953
75% Percentile	0.5456	0.5106	0.7589	0.9076	0.953
Maximum	0.5756	0.5306	0.7989	0.9276	0.973
Range	0.05	0.03	0.05	0.03	0.02
Mean	0.5468	0.5116	0.7614	0.9086	0.955
Std. Deviation	0.01012	0.005525	0.0102	0.005525	0.005231
Std. Error of Mean	0.002264	0.001235	0.00228	0.001235	0.00117
Sum	10.94	10.23	15.23	18.17	19.1

**Table 8 diagnostics-13-03439-t008:** The ANOVA test for the presented hyOPTGB model and other models.

	SS	DF	MS	F (DFn, DFd)	*p* Value
Treatment (between columns)	3.291	4	0.8227	F (4, 95) = 13,952	*p* < 0.0001
Residual (within columns)	0.005602	95	5.9 × 10^−5^	-	-
Total	3.297	99	-	-	-

**Table 9 diagnostics-13-03439-t009:** The Wilcoxon signed-rank test for the presented hyOPTGB model and other models.

	Random Forest	KNN	MLP	LSTM	hyOPTGB
Theoretical median	0	0	0	0	0
Actual median	0.5456	0.5106	0.7589	0.9076	0.953
Number of values	20	20	20	20	20
Wilcoxon signed-rank test					
Sum of signed ranks (W)	210	210	210	210	210
Sum of positive ranks	210	210	210	210	210
Sum of negative ranks	0	0	0	0	0
*p* value (two tailed)	<0.0001	<0.0001	<0.0001	<0.0001	<0.0001
Exact or estimate?	Exact	Exact	Exact	Exact	Exact
Significant (alpha = 0.05)?	Yes	Yes	Yes	Yes	Yes
How big is the discrepancy?					
Discrepancy	0.5456	0.5106	0.7589	0.9076	0.953

## Data Availability

Data is available up on request.

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
