# Peer review of "Optimizing HCV Disease Prediction in Egypt: The hyOPTGB Framework"

_diagnostics, 2023, doi:10.3390/diagnostics13223439_

Round 1
Reviewer 1 Report
Comments and Suggestions for Authors
1. Whether any literature study conducted in the classification and prediction of the Hepatitis C virus? If yes, the authors must include it in the literature part. line 129-130, It is mentioned that "Numerous studies have proposed various methods for predicting HCV disease". I suggest adding references here.
2. Can the authors include more basic information about the term "Hyperparameter" in the introduction part?
3. Whether any previous study conducted using OPTUNA? If yes, include it in the literature and justify your work with the previous research.
4. In the related work, Reference [32], the approach achieved 96.25% which is higher than the accuracy achieved by your proposed work reported. Can the authors justify the reason and also justify in what context the proposed work is better than that?
5. In Table 4, the comparison is made between the machine learning classifier models against your proposed work. In the proposed work initially the optimization of hyperparameters was done using OPTUNA. My question is whether the OPTUNA optimization was also applied for other classifiers (DT,SVM, etc.,).
6. The dataset information and its link should be given in the introduction part.
7. Tabel 6, add more previous work for comparison. Check ref [32]
8. Authors must include the novelty of the work clearly.
Comments on the Quality of English LanguageMinor editing of language is required
Author Response
Many thanks for your valuable comments and appreciate your support , all comments has been uploaded

Reviewer 2 Report
Comments and Suggestions for Authors
Authors have proposed an optimized gradient boosting classifier to predict HCV disease in Egypt. The proposed method has tuned hyperparameters and feature selection approaches to improve performance. Some major updates has to be done to improve the paper. Please find below the comments:
1) It is mentioned that the model is a classification model but in Table 7, RMSE values are calculated. This is confusing. Better to explain this part clearly.
2) Comparison of LSTM is not appropriate with other ML algorithms in this case. Does authors use validation plot?
3) Figure 9, 10, 11, 12 are not self explainable. More details are required
Author Response

(The authors gave the same response as above.)

Round 2
Reviewer 1 Report
Comments and Suggestions for Authors
In the revised manuscript, the authors have modified/included as per the suggestions given.
Reviewer 2 Report
Comments and Suggestions for Authors
Authors have addressed most of the comments and updated the paper. The paper can be accepted in the current format